# Oxidative Stress as a Contributor to Insulin Resistance in the Skeletal Muscles of Mice with Polycystic Ovary Syndrome

**DOI:** 10.3390/ijms231911384

**Published:** 2022-09-27

**Authors:** Qiyang Yao, Xin Zou, Shihe Liu, Haowen Wu, Qiyang Shen, Jihong Kang

**Affiliations:** Department of Physiology and Pathophysiology, School of Basic Medical Sciences, Peking University Health Science Center, No.38 Xueyuan Rd., Haidian District, Beijing 100191, China

**Keywords:** polycystic ovary syndrome, oxidative stress, insulin resistance, skeletal muscle

## Abstract

Polycystic ovarian syndrome (PCOS) is a reproductive, endocrine, and metabolic disorder. Circulating markers of oxidative stress are abnormal in women with PCOS. There is a close relationship between oxidative stress and insulin resistance (IR). However, little information is available about oxidative stress in the skeletal muscles of those affected by PCOS. In this study, PCOS was induced in prepubertal C57BL/6J mice by injection with dehydroepiandrosterone. Oxidative stress biomarkers were then measured in both serum and skeletal muscles. The underlying mechanisms were investigated in C2C12 myotubes treated with testosterone (T). We discovered increased oxidative biomarkers, increased ROS production, and damaged insulin sensitivity in the skeletal muscles of mice with PCOS. High levels of T caused mitochondrial dysfunction and increased ROS levels through the androgen receptor (AR)-nicotinamide adenine dinucleotide phosphate oxidase 4 (NOX4) signaling pathway in C2C12 cells. Treatment of C2C12 cells with an antioxidant N-acetylcysteine (NAC) decreased T-induced ROS production, improved mitochondrial function, and reversed IR. Administration of NAC to mice with PCOS improved insulin sensitivity in the skeletal muscles of the animals. Hyperandrogenism caused mitochondrial dysfunction and redox imbalance in the skeletal muscles of mice with PCOS. We discovered that oxidative stress contributed to skeletal muscle IR in PCOS. Reducing ROS levels may improve the insulin sensitivity of skeletal muscles in patients with PCOS.

## 1. Introduction

Polycystic ovary syndrome (PCOS) is an endocrine and reproductive disorder that affects 6% to 10% of reproductive-age women. Patients with PCOS experience the following symptoms: irregular menstruation, clinical or biochemical hyperandrogenism, or polycystic ovarian morphology on ultrasonography [1]. In most cases, PCOS also involves metabolic alterations, such as obesity, insulin resistance (IR), and dyslipidemia. Women with PCOS are at risk for increased long-term health complications, including metabolic syndrome, type 2 diabetes, non-alcoholic fatty liver disease (NAFLD), and cardiovascular diseases [2]. 

About 75% of these women experience impaired insulin sensitivity. Abnormalities of insulin action are present in classic insulin target tissues, such as skeletal muscle. Although several studies have demonstrated the relationship between IR and PCOS, the underlying mechanisms remain an unsolved issue. Skeletal muscle is responsible for approximately 80% of insulin-stimulated glucose uptake in the whole body. Therefore, skeletal muscle plays a critical role in peripheral glucose metabolism. The mechanisms involved in skeletal muscle insulin resistance include impaired glucose transporter 4 (GLUT4) translocation, mitochondrial dysfunction, lipotoxicity, increased production of reactive oxygen species (ROS), and so on. In PCOS, previous studies demonstrated the increased phosphorylation of insulin receptor substrate 1 (IRS1) [3], as well as reduced insulin-stimulated phosphorylation of protein kinase B (AKT) and its 160 kDa substrate (AS160) [4], in skeletal muscle. 

Oxidative stress is commonly used to represent an imbalance between the production of free radicals and the body’s ability to defend against their harmful effects through antioxidants, leading to DNA damage and apoptosis [5]. Interruptions in the normal cellular oxidation reaction and the production of free radicals and peroxides can lead to cytotoxic effects. ROS represent a class of molecules originating from oxygen metabolism in aerobic organisms. The main reactive oxygen species with physiological functions include: hydrogen peroxide (H_2_O_2_), superoxide anion (O_2_•−), hydroxyl radical (•OH), organic hydroperoxides (ROOH), alkoxyl and peroxyl radicals (RO and ROO), peroxynitrite (ONOO^−^), and hypochlorous acid (HOCl) [6]. ROS can be divided into endogenous and exogenous sources. In the human body, endogenous sources of ROS include: leakage from mitochondria during oxidative phosphorylation, leakage in detoxification of the hepatocellular cytochrome P450 enzyme system, peroxisomal oxidase, and NAD(P)H oxidase [7]. Exogenous sources include: alcohol consumption, exposure to environmental pollutants, cigarettes, ionizing radiation, etc [8]. 

Oxidative stress has been recognized in various pathological disorders related to IR. In the disease states associated with oxidative stress, the generation of ROS increases. It has been reported that oxidative stress plays a role in the pathophysiology of PCOS and is closely linked with hyperandrogenemia and IR [9]. Many studies have suggested that IR and hyperglycemia in women with PCOS are key factors in increasing oxidative stress [10]. In women with PCOS, evidence shows elevated ROS production and decreased mitochondrial mass in the ovarian granulosa cells [11]. Moreover, PCOS-like rodents exhibit increased ROS in ovaries [9] as well as leukocytes [12]. Recently, Li et al. reported that attenuating intestinal oxidative stress ameliorated PCOS phenotypes in a model rat with PCOS [13]. Whether oxidative stress contributes to IR in the skeletal muscles of patients with PCOS is unknown. 

In this study, we investigated oxidative stress and its possible role in skeletal muscle IR in DHEA-induced mouse models with PCOS. Our results demonstrated that hyperandrogenism caused mitochondrial dysfunction and redox imbalance in the skeletal muscle of mice with PCOS. ROS contributed to insulin resistance in the skeletal muscle of models with PCOS, which can be ameliorated by antioxidants such as N-acetylcysteine (NAC).

## 2. Results

### 2.1. Increased Oxidative Stress in the Whole-Body and Skeletal Muscles of Mice with DHEA-Induced PCOS

Oxidative-redox metabolomics analysis was performed in the serum and skeletal muscle tissues of the animals. Glutathione (GSH) is an endogenous tripeptide constituting the most abundant pool of cellular antioxidants. The ratio of GSH to its commonly observed oxidation product glutathione disulfide (GSSG) is an effective index for characterizing oxidative stress in a biological system. As shown in Figure 1A,B, there was a significant decrease in the ratio of GSH/GSSG in both serum and skeletal muscles of PCOS mice compared with control mice. The levels of ROS were also measured in the skeletal muscle samples. ROS levels in the soleus muscles of PCOS mice were higher than in those of the control animals (Figure 1C). These data suggested the increased oxidative stress in the whole-body and skeletal muscles of DHEA-induced PCOS mice. The expression of antioxidant genes was also measured in the skeletal muscles of the mice. In PCOS mice, the protein level of superoxide dismutase 2 (SOD2) in the skeletal muscles was significantly higher than in controls (Figure 2A,B). Moreover, the total SOD activity in the skeletal muscles was increased in mice with PCOS compared with control mice (Figure 2C). The expression of antioxidant genes *catalase* and *glutathione peroxidase 1* (*GPx1*) at the mRNA level was remarkably higher in the skeletal muscles of PCOS mice than in control mice (Figure 2D). There was no marked difference in the mRNA expression of antioxidant genes *SOD1*, *SOD3*, and *glutathione reductase* (*GR*) between PCOS mice and control mice. The mRNA levels of *NOX2* and *NOX4* were significantly increased in the skeletal muscles of PCOS mice compared with controls (Figure 2E). These results indicated the increased oxidative stress in the skeletal muscles of mice with PCOS.

### 2.2. Increased ROS Levels in Testosterone-Treated C2C12 Cells

To investigate the possible mechanisms underlying the increased oxidative stress in the skeletal muscles of PCOS mice, C2C12 cells were cultured and treated with testosterone (T). Fully differentiated C2C12 cells were treated with different concentrations of T (0, 10^−8^, 10^−7^, 10^−6^, 10^−5^, 10^−4^, and 10^−3^ M) for 12 h and 24 h. Then, cell viability was assessed by MTT assay. As shown in Figure 3A, there was no detectable change in cell viability when cells were treated with T at 10^−8^, 10^−7^, 10^−6^, and 10^−5^ M for 12 h or 24 h, as compared with the vehicle group (T at 0 M). The intracellular ROS levels were then quantified by DCFH-DA. There was an increase in the intracellular ROS in cells treated with T for 12 h in a dose-dependent manner (Figure 3B). Treatment of C2C12 cells with T at 10^−5^ M for 12 h was thus chosen in the subsequent experiments.

### 2.3. Impaired Mitochondrial Function and Insulin Sensitivity in T-Treated C2C12 Cells

One main source of cellular ROS is mitochondria. We thus determined the mitochondrial function of T-treated C2C12 cells. Previously, we found that treatment of C2C12 cells with T at 5 × 10^−7^ M for 24 h impaired mitochondrial function and induced insulin resistance [14]. In this study, we used T at 10^−5^ M to treat C2C12 cells for 12 h, and treatment with T at 0 M was used as a control (Ctrl). Mitochondrial membrane potential was measured. Our results showed that the ratio of JC-1 monomers to JC-1 aggregates was significantly lower in T-treated cells (Figure 4A,B), suggesting the decreased mitochondrial membrane potential in these cells. The major role of the mitochondria is to generate ATP for the cells. We then measured the ATP levels. The ATP production was markedly reduced in T-treated C2C12 cells, as shown in Figure 4C. Thereafter, we detected insulin signaling in C2C12 cells treated with 10^−5^ M of T for 12 h. Consistent with our previous report, cells treated with insulin+T exhibited blunted phosphorylation of Akt compared with insulin treatment alone (Figure 4D,E), suggesting the decreased insulin sensitivity in T-treated cells.

### 2.4. NOX4 Plays an Important Role in T-Induced ROS Production in C2C12 Cells

Handayaningsih et al. reported that NOX4 is dominantly expressed in C2C12 cells and it plays an important role in producing ROS in C2C12 cells [15]. We thus measured NOX4 expression in C2C12 cells treated with 10^−5^ M of T for 12 h. Cells treated with 0 M of T were used as control (Ctrl). As illustrated in Figure 5A, the relative mRNA level of *NOX4* was significantly increased in T-treated C2C12 cells compared with control. To figure out whether NOX4 plays a role in producing ROS in T-treated C2C12 cells, the expression of NOX4 was knocked down by transfection of siRNA-NOX4 in C2C12, followed by the treatment with 10^−5^ M of T for 12 h. The knockdown efficiency of *NOX4* in C2C12 cells was revealed by quantitative PCR (qPCR). The mRNA level of *NOX4* was reduced by about 42% with siRNA-NOX4 (Appendix A). We then examined the effect of NOX4 knockdown on ROS production in T-treated C2C12 cells. ROS production was significantly decreased when *NOX4* expression was knocked down in T-treated cells (Figure 5B). In addition, insulin sensitivity was improved by the knockdown of *NOX4* in T-treated C2C12 cells, as revealed by the marked difference in the ratio of p-AKT/AKT between the siRNA-NOX4+T+Insulin and the T+Insulin groups (Figure 5C,D). Mitochondrial membrane potential and ATP production were measured as well. As shown in Figure 5E,F, knockdown of *NOX4* increased T-treated cell mitochondrial membrane potential. Similarly, ATP production was significantly increased in the siRNA-NOX4+T group compared with the T group (Figure 5G). These results suggested that T induced ROS production in C2C12 cells by increasing *NOX4* expression.

### 2.5. T Induces ROS in C2C12 Cells by Binding to the Androgen Receptor (AR)

To explore the mechanism by which high concentrations of androgens induced ROS in the skeletal muscle of mice with PCOS and C2C12 cells, we first measured the expression of AR. AR expression in skeletal muscle was significantly increased in PCOS mice compared with controls (Figure 6A). Consistently, the expression of AR was markedly higher in T-treated C2C12 cells than in control cells (Figure 6B). These data suggested the involvement of AR in the pathophysiology of skeletal muscles of mice with PCOS. 

To further investigate whether T induced ROS through AR, the levels of AR were knocked down by siRNA interference. Three sequences targeting *AR* mRNA and one scrambled nonsilencing RNA were constructed and successfully transfected into C2C12 cells for 36 h. Cells transfected with scrambled nonsilencing RNA were used as control (NC). The efficiency of siRNA-AR knockdown was revealed by qPCR, and the results showed that the mRNA levels of *AR* by siRNA3-AR were reduced to less than half (Appendix A). Therefore, siRNA3-AR was chosen and utilized in the subsequent experiments. AR is a classical ligand-activated nuclear receptor. We thus first studied AR subcellular localization and expression after siRNA-AR transfection by using confocal microscopy. As illustrated in Figure 6C, the AR signal was predominantly extranuclear in the absence of T in C2C12 cells. Upon T stimulation, AR underwent nuclear translocation. The transfection of siRNA-AR decreased the AR signal both inside and outside of the nuclei. An androgen receptor antagonist flutamide was also used. Flutamide blocked part of AR translocation into nuclei upon T treatment (Appendix A). ROS levels were then detected. Fully differentiated C2C12 cells were transfected with siRNA-AR for 36 h, followed by T (10^−5^ M) treatment for 12 h. ROS levels induced by T were significantly decreased in AR-knockdown C2C12 cells (Figure 6D). Similarly, flutamide reduced T-induced ROS production as well (Appendix A). These results suggested that T induced ROS in C2C12 cells through binding to AR. 

We next examined *NOX4* expression and cell insulin signaling after AR knockdown in C2C12 cells. Knockdown of AR significantly reduced the mRNA expression of T-induced NOX4 (Figure 6E). Consistently, the ratio of p-AKT/AKT was increased with the treatment of siRNA-AR+T+Insulin compared with T+Insulin, indicating the upregulated insulin sensitivity in the cells when AR expression was knocked down (Figure 6F,G). 

In summary, these data suggested that T induced ROS and reduced insulin sensitivity in C2C12 cells.

### 2.6. Reducing ROS Levels Ameliorates Mitochondrial Function and Insulin Resistance in T-Treated C2C12 Cells

To investigate whether reducing ROS could ameliorate mitochondrial function and cell insulin resistance, an antioxidant NAC was used. Fully differentiated C2C12 cells were treated with T (10^−5^ M) or T (10^−5^ M) plus NAC (1, 2, and 10 mM) for 12 h. ROS production was then measured. As shown in Figure 7A, ROS levels were decreased in cells treated with T+NAC in a dose-dependent manner compared with T treatment alone. In the subsequent experiments, 1 mM of NAC was then used to treat the C2C12 cells in combination with T. The cotreatment of NAC with T significantly upregulated ATP levels and mitochondrial membrane potential (Figure 7B,D), indicating the amelioration of mitochondrial function in T-treated C2C12 cells by reducing ROS levels. In addition, cell insulin sensitivity was measured. The increased ratio of p-AKT/AKT in C2C12 cells treated with T+NAC+Insulin in comparison with T+Insulin treatment suggested the increased insulin sensitivity in T-treated cells (Figure 7E,F). In general, these results suggest that reducing ROS by the treatment with antioxidants such as NAC improved T-induced mitochondrial function and ameliorated insulin resistance in C2C12 cells.

### 2.7. NAC Ameliorates Insulin Resistance in the Skeletal Muscle of DHEA-Induced PCOS Mice

To further confirm the effect of antioxidants on skeletal muscle insulin resistance in mice with PCOS, NAC was given to DHEA-induced PCOS mice when modeling. ROS production was measured in soleus muscle tissues. ROS levels were attenuated in PCOS+NAC mice (Figure 8A). Additionally, the insulin sensitivity of skeletal muscle was detected. The blunted phosphorylation of AKT in PCOS mice was rescued by NAC intervention, as shown in Figure 8B,C. These results suggested that reducing ROS levels by antioxidants could improve skeletal muscle insulin sensitivity in mice with PCOS.

## 3. Discussion

Many studies have revealed that women with PCOS have higher circulating oxidative markers compared with healthy controls [16]. ROS accumulates in the follicular fluid of women with PCOS [17]. In response to saturated fat ingestion, PCOS patients exhibit increased leukocytic ROS generation independent of obesity [18]. Consistent with human studies, results from animal models also showed oxidative stress in PCOS rodent models. Mujica and colleagues found that oxidative stress was promoted in testosterone-induced PCOS rats [19]. In agreement with these studies, our data revealed that high levels of androgens increased whole-body oxidative stress, both in DHEA-induced PCOS mice and T-treated C2C12 cells. 

Oxidative stress is an imbalance derived from excessive oxidants in the presence of limited antioxidant defenses. The production and propagation of intracellular ROS are controlled by antioxidant enzymatic and non-enzymatic systems. ROS comes from the mitochondria and other cellular sites [7]. One source of ROS is the nicotinamide adenine dinucleotide phosphate (NADPH) oxidases (NOX) family, which consists of seven members, including NOX1-5 and two dual oxidases (Duox), Duox1 and Duox2 [20,21]. NOX-mediated ROS plays an important role in cell signaling and cell differentiation [22]. In C2C12 cells, it has been shown that NOX4 is the dominated isoform [23,24]. 

SOD, catalase (CAT), and GPx play a key role in defending against oxidative stress. SOD eliminates superoxide anions by catalyzing them to H_2_O_2_, which is finally converted to water by GPx. Previous studies have found that women with PCOS have elevated blood SOD levels compared with healthy controls [25]. Data from a meta-analysis also showed that the mean SOD activity was 34% higher in PCOS patients than in controls [16]. Consistent with these results, we observed the increased SOD activity and SOD2 expression in skeletal muscles of mice with PCOS in our study. However, a study by Mohammadi and colleagues showed that the serum SOD level in PCOS patients was significantly lower than that in controls [26]. Therefore, further studies are needed to determine the role of SOD in the pathology of PCOS. GSH is an important cellular antioxidant and is distributed in the endoplasmic reticulum, nucleus, and mitochondria. Dincer et al. have reported that GSH levels were significantly lower in women with PCOS than in the controls [27]. They proposed that GSH depletion may be due to the increased ROS production in PCOS patients. Similarly, Sabuncu and colleagues also found decreased GSH levels in PCOS women [28]. In our study, we found a significantly decreased ratio of GSH/GSSG in serum and skeletal muscle in mice with PCOS compared with control mice, indicating the increased systemic and skeletal muscle ROS levels in PCOS. 

Oxidative stress is now considered to play an important role in the pathogenesis of PCOS. Characteristics of PCOS, such as androgen excess, abdominal adiposity, insulin resistance, and obesity, may contribute to the development of systemic oxidative stress, while oxidative stress may reciprocally worsen these metabolic abnormalities. Women with PCOS exhibit a generalized IR, as well as a blunt response to insulin in adipose tissue and skeletal muscle [29]. Traditionally, ROS is regarded as a toxic byproduct of metabolism that can cause organ dysfunction [30]. It is reported that ROS is likely to play a role in IR. In type 2 diabetes, IR is thought to result in chronic ROS produced by mitochondria [31,32]. Several pharmacological interventions, including NAC, MnTBAP, and knocking in encoding ROS-scavenging enzymes genetic intervention, decrease ROS levels to prevent IR in 3T3-L1 cells [33]. In the present study, we found increased oxidative stress in the skeletal muscle of mice with PCOS. NAC, a metabolite of the sulfur-containing amino acid cysteine, is currently used as an antioxidant and mucilage dissolver. The therapeutic potential of NAC has been investigated for a range of disease treatments, such as antidotes to specific toxins, bioprotectants against oxidative stress and ischemic injury, and therapeutic agents for certain mental and physical disorders. NAC is a scavenger of reactive oxygen species. As a precursor of cysteine, it possesses an important role in a rate-limiting step in glutathione synthesis. Depletion of GSH under oxidative stress can be reversed by NAC supplementation [34]. It has been reported that NAC can increase the peripheral insulin sensitivity in hyperinsulinemic patients with PCOS [35,36]. After NAC administration, hyperinsulinemic women with PCOS showed an increase in glucose utilization and a significant reduction in insulin AUC [37]. NAC was thus tested in our experiments. As expected, administration of NAC to PCOS mice or the cotreatment of C2C12 cells with T+NAC reduced ROS levels and ameliorated IR, possibly through ameliorating mitochondrial function and reducing ROS both in vivo and in vitro. The relationship between hyperandrogenism and insulin resistance in PCOS is still controversial. Hyperinsulinemia contributes to hyperandrogenemia [29]. However, some studies hypothesized that PCOS is the vicious circle of androgen excess by inducing insulin resistance and compensatory hyperinsulinism, which further facilitates androgen secretion in women with PCOS [38]. Our data confirmed that high levels of androgens induce oxidative stress that contributes to skeletal muscle IR in PCOS. 

Among all androgens, testosterone as well as dihydrotestosterone is more active than others. Testosterone exerts its actions by binding to AR, which is a member of the nuclear receptor family. Upon testosterone binding, AR signals through both genomic and non-genomic mechanisms [39,40]. The classical function model of testosterone is mediated by AR localized in the nucleus, whereas a study showed that T aggravates cytotoxicity in an oxidative stress environment independent of AR [41]. Pronsato et al. showed a non-classical distribution of membrane AR that can respond to testosterone treatment in C2C12 cells [42]. T binds to AR and can induce migration of vascular smooth muscle cells (VSMCs) via NADPH oxidase-driven ROS in WKY rats and SHR VSMSs through both genomic and non-genomic pathways [43]. We thus determined whether androgens induced oxidative stress via AR. Our data showed the increased AR expression in the skeletal muscle of mice with PCOS. Additionally, results of AR subcellular localization by confocal microscopy, the effects of AR-knockdown by siRNA transfection, and the effects of AR blocker flutamide on ROS and insulin sensitivity in T-treated C2C12 cells suggested that androgens induced oxidative stress via AR through the genomic pathway.

A limitation of our study lies in that the in vitro mechanism experiments were performed in mouse C2C12 cells. Due to the complex structural arrangement of the skeletal muscle, mouse C2C12 cells represent the most used cellular models to study skeletal muscle in vitro. Evidence shows that C2C12 cells can be considered an appropriate model to investigate skeletal muscle metabolism because C2C12 cells can undergo differentiation into elongated myotubes and express the insulin-responsive GLUT4 protein [44]. However, it is still possible that data from these cells could be different from that of skeletal muscles in vivo. 

In summary, our data revealed that high levels of androgens induced oxidative stress and increased ROS levels. Oxidative stress contributed to skeletal muscle insulin resistance in PCOS. Reducing ROS levels may improve the insulin sensitivity of skeletal muscle in PCOS patients.

## 4. Materials and Methods

### 4.1. Animals and Treatments

Female C57BL/6J mice aged 21 days were purchased from Beijing HFK Bio-Technology. Co., LTD (Beijing, China). Mice were housed at 22 ± 2 °C under the standard laboratory conditions (12L:12D cycle) with free access to rodent feed and water. At postnatal day 25, the mice were randomly divided into two groups with comparable body weight. Group 1: control group. Group 2: DHEA-induced PCOS group. The control group and PCOS groups were treated as described previously [11]. In brief, the control mice were injected (s.c.) daily with sesame oil (0.1 mL per 100 g body weight) and the PCOS mice were injected (s.c.) daily with DHEA (6 mg per 100 g body weight) dissolved in 0.1 mL of sesame oil. After 20 days of treatments, the mice were sacrificed, and tissue samples were collected. For the rescue experiments, there were four groups of animals. Group 1: control group. The mice were injected (s.c.) daily with sesame oil (0.1 mL/100 g body weight). Group 2: PCOS group. Mice were injected (s.c.) daily with DHEA (6 mg/100 g body weight) dissolved in 0.1 mL of sesame oil. Group 3: NAC group. The mice were administered NAC (2g/L) dissolved in drinking water and injected (s.c.) daily with sesame oil (0.1 mL/100 g body weight). Group 4: PCOS+NAC group. Mice were administered NAC (2g/L) in drinking water and injected (s.c.) daily with DHEA (6 mg per 100 g body weight) dissolved in 0.1 mL of sesame oil. The mice were treated for 20 days and then were sacrificed. The blood and skeletal muscle samples were collected. All animal protocols were approved by the Animal Care and Use Review Committee of Peking University Health Science Center under the Guide for the Care and Use of Laboratory Animals published by the US National Institutes of Health. DHEA, sesame oil, and NAC were purchased from Sigma-Aldrich (St-Louis, MO, USA).

### 4.2. Tissue Collection

After 20 days of treatments, the mice were deeply anesthetized and harvested. Blood samples were collected via medial canthus puncturing. Skeletal muscles from the hind limbs, including soleus muscles, gastrocnemius, and quadriceps muscles, were collected for oxidative-redox metabolomics analysis, detection of SOD activity, Western blot, and RNA extraction. The soleus muscles were also used for ROS detection. 

### 4.3. Oxidative-Redox Metabolome in Serum and Tissues

The samples were prepared as described [45]. In brief, 100 μL of serum samples were mixed with 80% aqueous methanol (*v/v*) and equilibrated at −80 °C for 20 min. For tissues, 800 μL of extraction solvent (methanol/water 4:1 *v/v*, 20 μg/mL of 2-chloroadenosine) was added to 20 mg of tissues. The sample was centrifuged at 12,000× *g* for 10 min. The supernatants were then collected and centrifuged again after equilibration at −80 °C for 20 min. The supernatants were collected, dried, and stored at −80 °C. Samples were redissolved in 120 μL of deionized water and centrifuged at 15,000× *g* for 10 min. 100 μL of supernatant was collected and 400 μL of methanol containing internal standard was added for injection. Chromatographic separation was performed with a UHPLC system (Waters Corp., Milford, MA, USA) with a Waters Xbridge amide column (100 × 4.6 mm, 3.5 μm). The mobile phase consisted of A (5 mM ammonium acetate in water) and B (100% acetonitrile, ACN) with a flow rate of 0.5 mL/min. The MS system was operated using the QTRAP 5500 (AB SCIEX, Concord, ON, Canada). The data acquisition and processing were carried out using Analyst Software (AB SCIEX) and MultiQuant Software (AB SCIEX).

### 4.4. Cell Culture and Treatments

Mouse C2C12 myoblasts were purchased from China Infrastructure of Cell Line Resource (Beijing, China). Cells were cultured in Dulbecco’s Modified Eagle’s Medium (DMEM) containing 4500 mg/L glucose and supplemented with 10% fetal bovine serum (FBS), 4 mM glutamine, and 100 U/mL penicillin/streptomycin, at 37 °C in a humidified atmosphere with 5% CO_2_. The culture medium was changed every 2 days. At 80% confluence, the cells were allowed to differentiate by replacing growth media with DMEM supplemented with 2% horse serum and 100 U/mL penicillin/streptomycin for 5 days. The fully differentiated C2C12 cells were treated with testosterone (Sigma-Aldrich) dissolved in alcohol, in the non-serum DMEM medium with 100 U/mL penicillin/streptomycin, for different periods.

### 4.5. Detection of ROS and Total Superoxide Dismutase (SOD) Activity

Frozen soleus muscle sections (10-μm thickness) were processed for the measurement of ROS using an ROS detection kit (GENMED, ARLINGTON, MA, USA) with 2′,7′-dichlorodihydrofluorescein diacetate (DCFH-DA). Fluorescent signals were observed under a fluorescence microscope. For in vitro experiments, intracellular ROS was also detected in cells by DCFH-DA (Beyotime Institute of Biotechnology, Shanghai, China). In brief, a total of 2 × 10^6^ fully differentiated C2C12 cells were treated with testosterone (0, 10^−5^ M) for 12 h. Then, the cells were washed with PBS and incubated with DCFH-DA staining working solution for 20 min at 37 °C. The solution was then removed, and cells were washed twice with PBS. The fluorescence of the cells was monitored immediately using a fluorescence microscope. Total SOD activity in skeletal muscles was detected with a commercially available kit (Beyotime Institute of Biotechnology). 

### 4.6. Determination of ATP Content

ATP contents of cultured cells were measured using the ATP-Lite Assay Kit (Vigorous Biotechnology, Beijing, China). The ATP content was measured using a Luminometer (Turner BioSystems, Sunnyvale, CA, USA), normalized by the protein concentration (nmol/mg protein) in the same sample, and presented as the fold of the control.

### 4.7. Mitochondrial Membrane Potential Determination

The mitochondrial membrane potential of the cultured cells was detected with the mitochondrial membrane potential assay kit with JC-1 (Beyotime Institute of Biotechnology). A total of 2 × 10^6^ fully differentiated C2C12 cells were treated with testosterone (0, 10^−5^ M) for 12 h. The cells were then incubated with the JC-1 staining working solution for 20 min at 37 °C. The solution was then removed, and the cells were washed twice with the buffer solution. The fluorescence of the cells was monitored immediately using a fluorescence microscope.

### 4.8. Western Blot Analysis

Skeletal muscle samples or cultured cells were homogenized in lysis buffer and quantified as previously described [11]. The proteins of samples were separated by 12% SDS-PAGE and transferred to nitrocellulose membranes. The membranes were incubated for 2 h at room temperature with 5% fat-free milk in Tris-buffered saline containing Tween 20, followed by incubation at 4 °C with primary antibodies overnight. The antibodies to AKT (9272S) and phosphor-AKT at Ser473 (p-AKT) (4060S) were purchased from Cell Signaling Technology. GAPDH (ZAGB, China) was used as an internal control of total protein. The antibody to SOD2 was purchased from Bioss, Beijing, China. The antibody to the androgen receptor (1:1000) was bought from Abcam, UK. The membranes were washed with TBS-T buffer 5 times and then incubated with a horseradish peroxide-conjugated secondary antibody for 1 h at room temperature. After washing, the membrane was developed with ECL Reagent (Merck Millipore, Billerica, MA, USA) and visualized using an enhanced chemiluminescence detection system (Tanon, Shanghai, China). The density of the signals was quantified with Image J (Version 1.53a 4, NIH) software.

### 4.9. Extraction of RNA and Quantitative Real-Time PCR Analysis

Total RNA was isolated from mouse skeletal muscle samples or the cultured C2C12 cells using RNAiso Plus (TaKaRa, Tokyo, Japan). Reverse transcription and quantitative real-time PCR were performed as previously described [11]. PCR analysis was performed in duplicates, and each experiment was repeated at least three times. The expression of the target genes was normalized to that of *GAPDH* in the same sample using the 2^−ΔΔCt^ method. The primer sequences were listed in Appendix A.

### 4.10. Small Interfering RNA (siRNA) Transfection

Three siRNAs containing sequences targeting the mouse *AR* or *NOX4* mRNA and one non-silencing RNA as negative control were obtained from HIPPO Biotechnology, China. The oligo sequences of *AR* and *NOX4* siRNA were shown in Appendix A. In brief, C2C12 cells were seeded and cultured in DMEM with 2% horse serum. The siRNAs or scrambled non-silencing RNA (NC) diluted in the transfection medium were transfected into C2C12 cells using the transfection reagent Lipo8000 (Beyotime Institute of Biotechnology) according to the manufacturer’s instructions. The cells were kept in the transfection medium for 12 h and the siRNA-containing medium was then changed to DMEM with 2% horse serum and 100 U/mL penicillin/streptomycin. Thirty-six hours after transfection, the cells were either harvested to validate the transfection efficiency or were utilized for other experiments.

### 4.11. Immunofluorescence Microscopy Analysis

Fully differentiated C2C12 cells were grown on chamber slides. After the treatments, cells were fixed in 4% paraformaldehyde and then permeabilized in 0.3% Triton-X (Sigma-Aldrich). Cells were blocked in 5% bovine serum albumin for 1 h at room temperature and then incubated with rabbit monoclonal antibody against androgen receptor (Abcam, Cambridge, UK) at a dilution of 1:200. Immunofluorescence was obtained by reaction with a fluorescent secondary antibody (Alexa 555 goat anti-rabbit secondary antibody, Bioss, Beijing, China) at a dilution of 1:250 for 1 h at room temperature. Slides were then rinsed three times with PBS. Fluorescence was visualized using a fluorescence microscope (Leica, Wetzlar, Germany).

### 4.12. Insulin-Signaling Assay in Animals and Cultured Cells

Fully differentiated C2C12 cells were treated with T (0, 10^−5^ M) for 12 h, followed by the treatment with insulin (0, 100 nM) for 5 min. Then, the cells were collected for protein assay as previously described [11]. For the animals, after the treatments for 20 days, three mice from each group were fasted overnight and then injected intraperitoneally with 1.0 IU/kg body weight of insulin or 0.9% saline. Skeletal muscles from the hind limbs (quadriceps muscle) were collected 15 min after the injection and snap frozen in liquid nitrogen for protein extraction. The protein expression of total AKT and phosphorylated AKT (Ser473) was determined by Western blot analysis.

### 4.13. MTT Assay

The MTT assay was used to evaluate the viability of the cells. The assay was performed according to a protocol from Cold Spring Harbor [46]. In brief, cells were planted in a 96-well plate. After treatments, the medium was removed and replaced with 100 μL of fresh culture medium. Then, 10 μL of the 12 mM MTT (Sigma-Aldrich) stock solution was added into each well, including a negative control. Cells were incubated for 4 h at 37 °C and DMSO (Sigma-Aldrich) was used to solubilize the formazan. Samples were mixed, and the absorbance was then measured at 490 nm using an ELISA reader (Thermo Scientific, Waltham, MA, USA).

### 4.14. Statistical Analysis

Data are presented as mean ± SEM. Statistical analysis was performed using GraphPad Prism software (Version IX; La Jolla, CA, USA). The D’Agostino-Pearson omnibus and Shapiro-Wilk tests were applied to test the normal distribution of values. Effects of the treatments were analyzed by an unpaired t-test for comparisons between two groups and by one-way analysis of variance (ANOVA), followed by Bonferroni’s posttest for multiple comparisons. *p* < 0.05 was considered statistically significant.

## Figures and Tables

**Figure 1 ijms-23-11384-f001:**
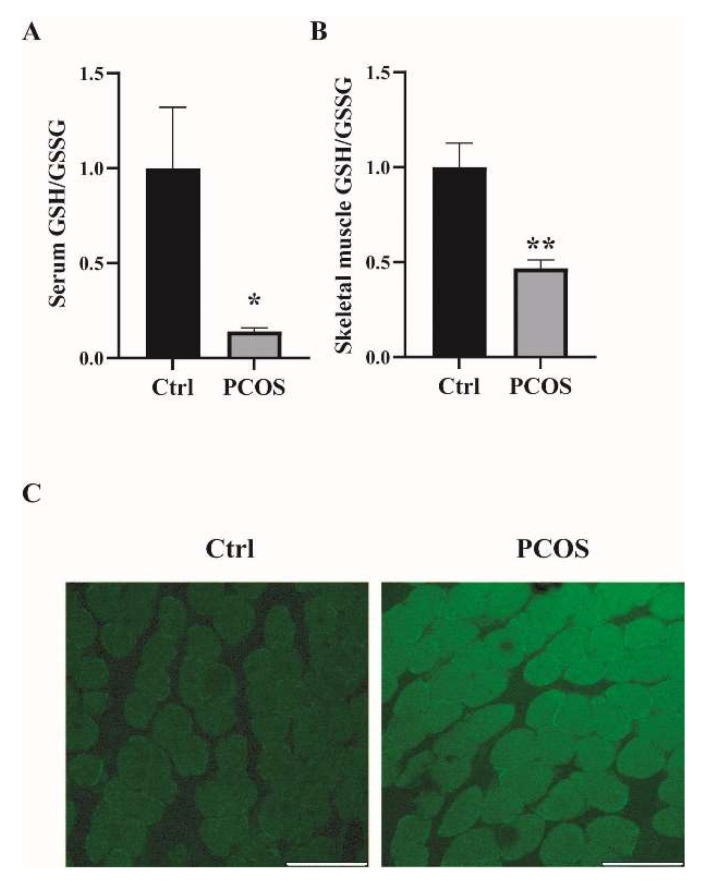
Increased oxidative stress in the whole-body and skeletal muscles of mice with DHEA-induced PCOS. (**A**) Metabolomics analysis of GSH/GSSG in serums of PCOS and control mice. (**B**) Metabolomics analysis of GSH/GSSG in the skeletal muscles of PCOS and control mice. (**C**) Representative photomicrographs of ROS relative fluorescence units (RFU) in the soleus muscles were measured by staining with DCFH-DA. Scale bar = 100 μm. Asterisks indicate significant differences between control and PCOS mice. The data are presented as means ± SEM. *, *p* < 0.05, **, *p* < 0.01, *n* = 6,7 per group.

**Figure 2 ijms-23-11384-f002:**
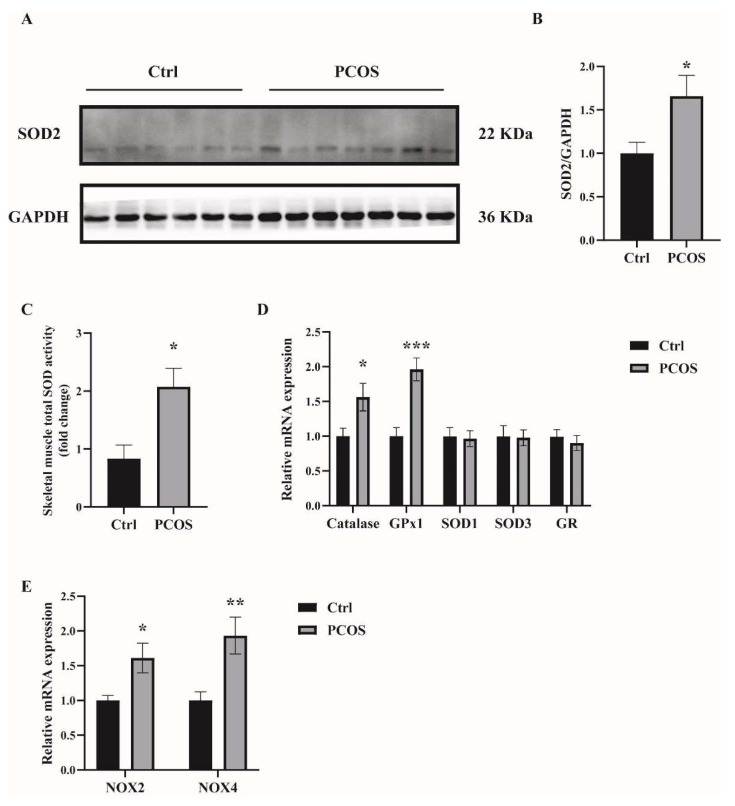
The expression of antioxidant-related genes in the skeletal muscles of the mice. (**A**,**B**) Western blot analysis and densitometry quantification of SOD2 in skeletal muscles of PCOS and control mice. (**C**) Total SOD activity in the skeletal muscles of PCOS and control mice. (**D**) Relative mRNA levels of the genes *Catalase*, *GPx1*, *SOD1*, *SOD3*, and *GR* in the skeletal muscles of PCOS and control mice. (**E**) Relative mRNA levels of the genes *NOX2* and *NOX4* in the skeletal muscles of PCOS and control mice. Asterisks indicate significant differences between control and PCOS mice. The data are presented as means ± SEM. *, *p* < 0.05, **, *p* < 0.01, ***, *p* < 0.001, *n* = 6,7 per group.

**Figure 3 ijms-23-11384-f003:**
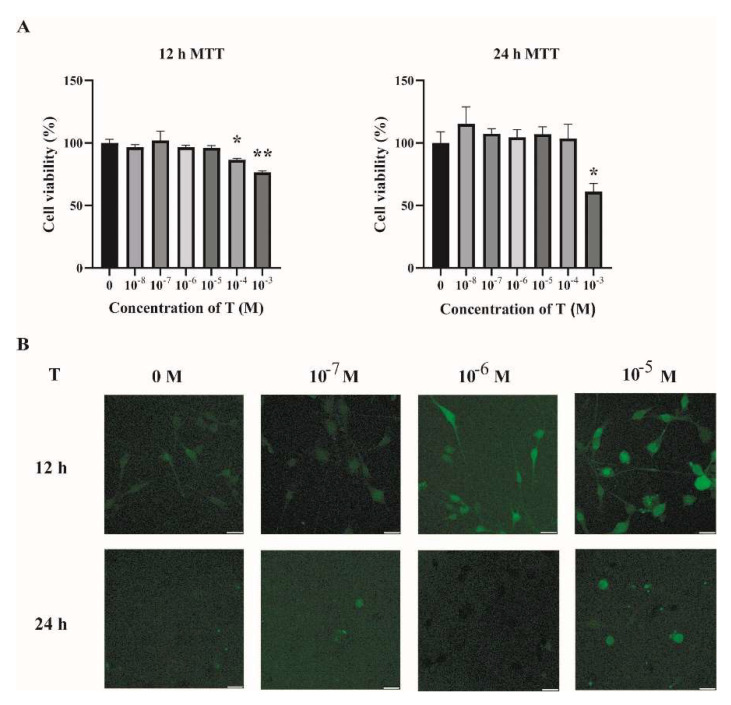
Increased ROS levels in T-treated C2C12 cells. (**A**) Fully differentiated C2C12 cells were treated with T (0, 10^−8^, 10^−7^, 10^−6^, 10^−5^, 10^−4^, and 10^−3^ M) for 12 h and 24 h. Cell viability was then measured by MTT assay. The data are means ± SEM of three independent experiments. Asterisks indicate significant differences between the vehicle-treated group and the different T treatment concentrations. *, *p* < 0.05, **, *p* < 0.01. (**B**) Representative photomicrographs of intracellular ROS detection in C2C12 cells after exposure to T at different concentrations for 12 h and 24 h. Intracellular ROS levels were quantified by DCFH-DA staining. Scale bar = 25 μm. T= testosterone.

**Figure 4 ijms-23-11384-f004:**
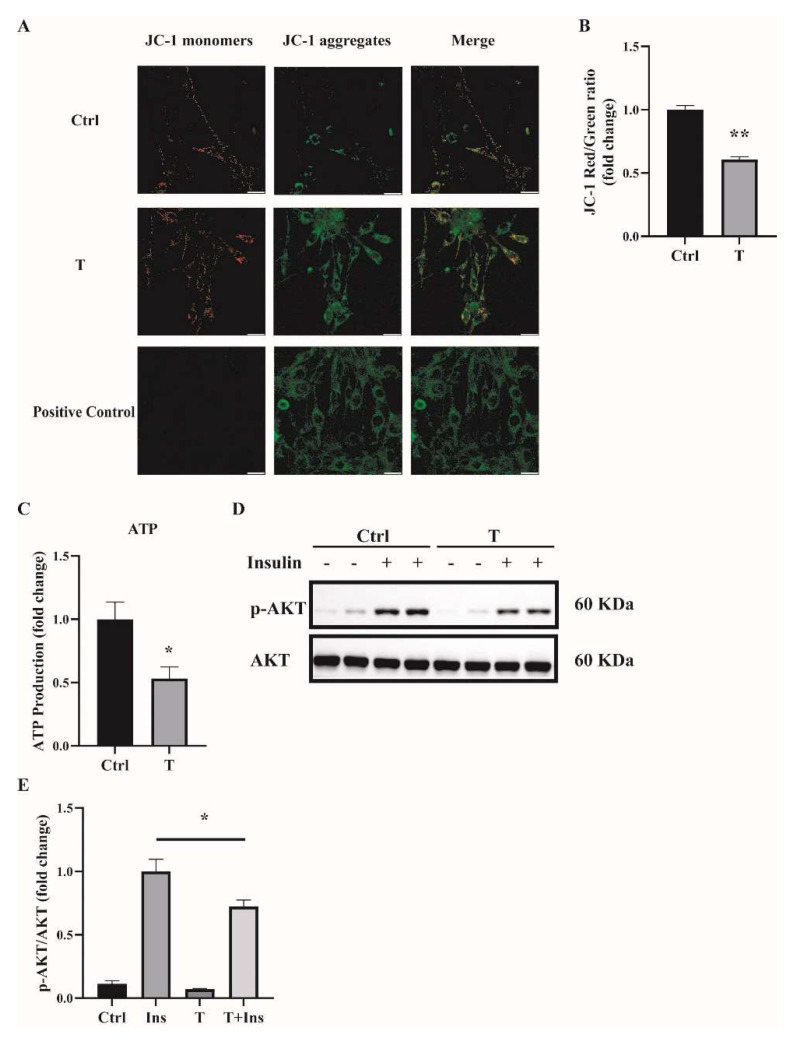
Impaired mitochondrial membrane potential, mitochondrial function, and insulin sensitivity in T-treated C2C12 cells. Fully differentiated C2C12 cells were treated with T (0 and 10^−5^ M) for 12 h. (**A**) The mitochondrial membrane potential (ΔΨm) was determined by JC-1. Cells treated with CCCP (10 μm) for 20 min were used as the positive control. The signals were observed using fluorescence microscopy. Representative images of three independent experiments are shown. The red fluorescence represents the mitochondrial aggregate form of JC-1, indicating high ΔΨm. The green fluorescence represents the monomeric form of JC-1, indicating the dissipation of ΔΨm. Scale bar = 25 μm. (**B**) The ratios of JC-1 aggregate/monomer were analyzed by the Image J program. The reduced ratio of JC-1 red/green indicated the damaged mitochondrial membrane potential. (**C**) Normalized ATP production. (**D**) Insulin-signaling assay. (**E**) Densitometry quantification of pAKT/AKT of the insulin-signaling assay in T-treated C2C12 cells. The data are presented as means ± SEM of three independent experiments. *, *p* < 0.05, **, *p* < 0.01. T, testosterone; Ins, insulin.

**Figure 5 ijms-23-11384-f005:**
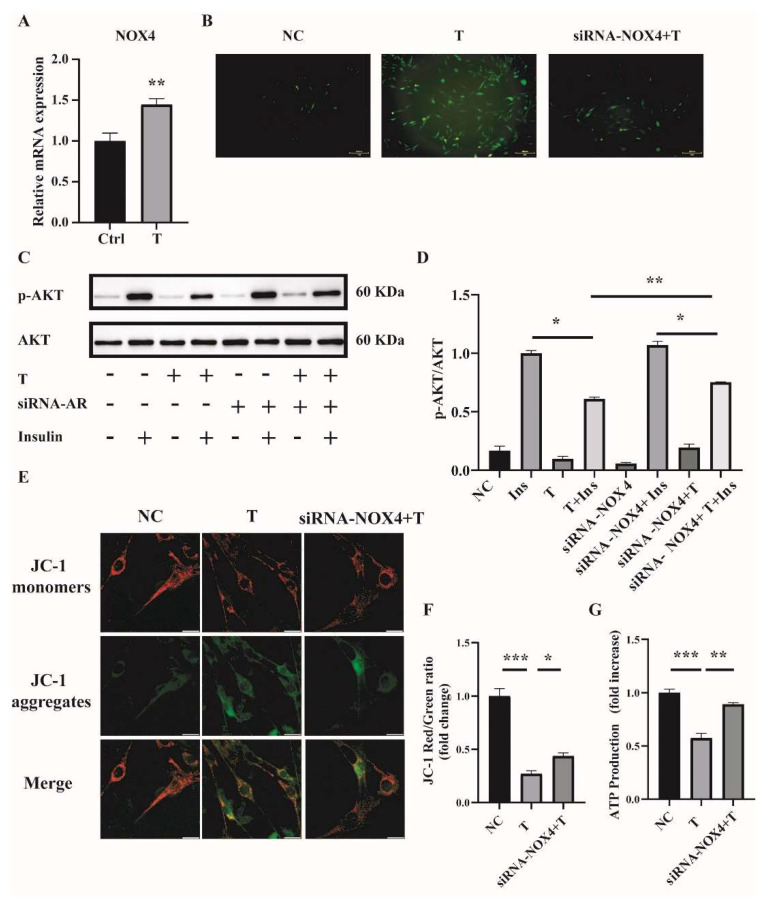
*NOX4* expression in T-treated C2C12 cells and effects of NOX4 on mitochondrial function. (**A**) Fully differentiated C2C12 cells were treated with T (0 and 10^−5^ M) for 12 h. Relative mRNA levels of *NOX4* in the cells. (**B**) Representative photomicrographs of intracellular ROS in C2C12 cells. Fully differentiated C2C12 cells were transfected with siRNA-NOX4 or non-silencing RNA and then treated with T (0, 10 μM) for 12 h. Intracellular ROS levels were determined by DCFH-DA staining. Scale bar= 200 μm(**C**) Insulin-signaling assay. (**D**) Densitometry quantification of pAKT/AKT of the insulin-signaling assay in C2C12 cells. (**E**) The mitochondrial membrane potential (ΔΨm) was determined by JC-1. Scale bar = 25 μm. (**F**) The ratios of JC-1 aggregate/monomer were analyzed by the Image J program. (**G**) Normalized ATP production. The data are presented as means ± SEM of three independent experiments. *, *p* < 0.05, **, *p* < 0.01, ***, *p* < 0.001. T, testosterone; Ins, insulin.

**Figure 6 ijms-23-11384-f006:**
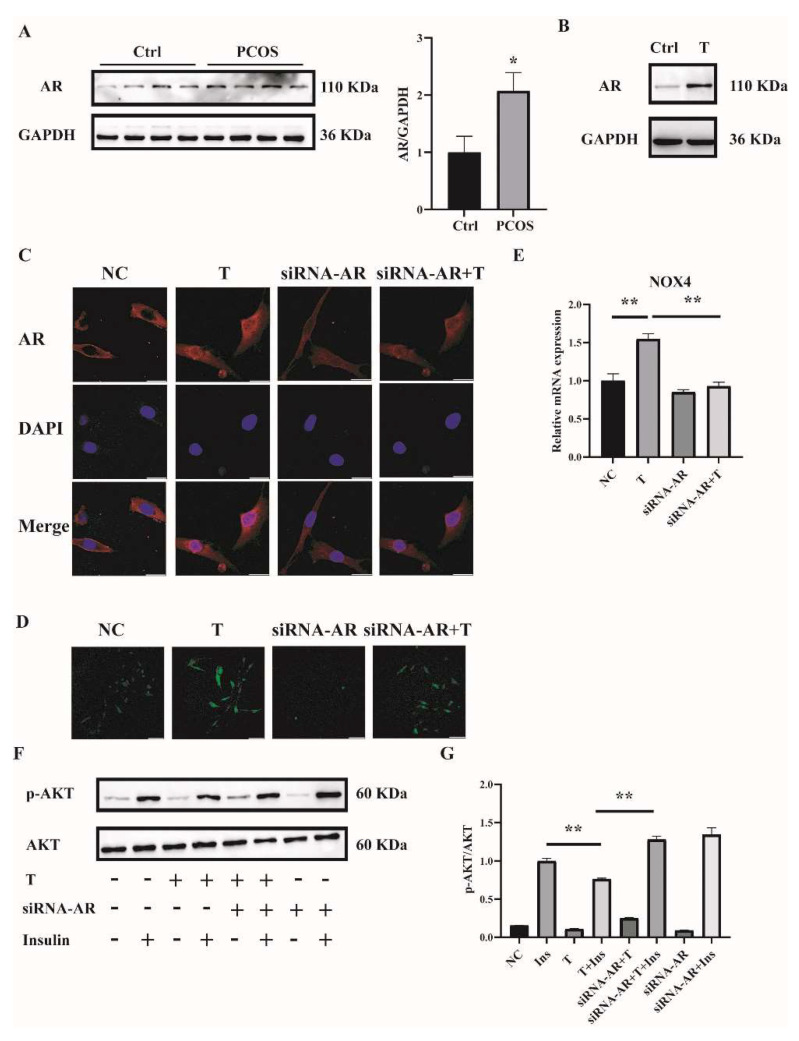
AR expression in the skeletal muscle of mice with PCOS and the effects of AR deficiency on the cellular oxidative stress in T-treated C2C12 cells. (**A**) Representative Western blots and densitometry quantification of AR in skeletal muscle of PCOS and control mice. (**B**) Representative Western blots of AR expression of T-treated C2C12 cells. Fully differentiated C2C12 cells were treated with T (0, 10 μM) for 12 h. AR expression was measured by Western blot analysis. (**C**) Representative photomicrographs of the subcellular localization of AR in C2C12 cells. Fully differentiated C2C12 cells were transfected with siRNA-AR or nonsilencing RNA (NC) and then treated with T (0, 10 μM) for 12 h. Red: AR; Blue: DAPI. Scale bar = 25 μm. (**D**) Representative photomicrographs of intracellular ROS in C2C12 cells. Fully differentiated C2C12 cells were transfected with AR siRNA or non-silencing RNA and then treated with T (0, 10 μM) for 12 h. Intracellular ROS levels were determined by DCFH-DA staining. Sale bar = 100 μm. (**E**) Relative mRNA levels of *NOX4* in T-treated C2C12 cells transfected with siRNA-AR non-silencing RNA (NC). (**F**) Insulin-signaling assay. (**G**) Densitometry quantification of pAKT/AKT of the insulin-signaling assay in C2C12 cells. The data are presented as means ± SEM of three independent experiments. *, *p* < 0.05, **, *p* < 0.01. T, testosterone; AR, androgen receptor; Ins, insulin.

**Figure 7 ijms-23-11384-f007:**
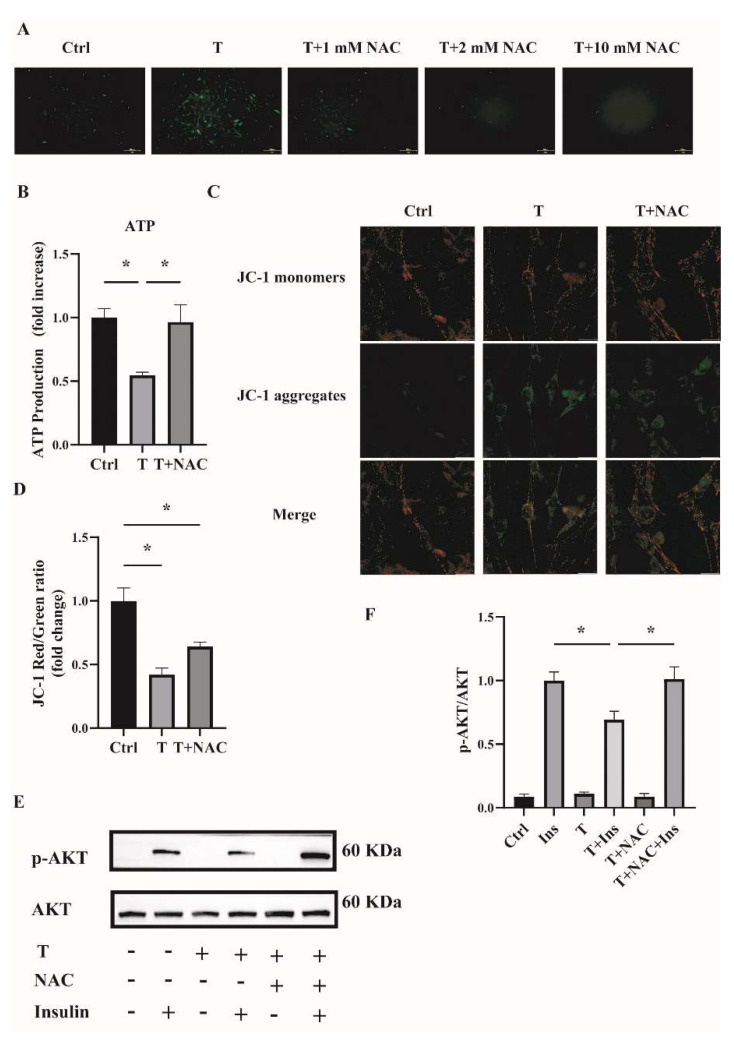
Effects of NAC intervention on T-treated C2C12 cells. (**A**) Representative photomicrographs of intracellular ROS in C2C12 cells. Fully differentiated C2C12 cells were treated with T (0, 10^−5^ M) or the combination of T (10^−5^ M) with NAC (1, 2, and 10 mM) for 12 h. Then, intracellular ROS was determined by DCFH-DA staining. Scale bar= 200 μm. (**B**) Normalized ATP production. Fully differentiated C2C12 cells were treated with T (0, 10^−5^ M) or T (10^−5^ M) plus NAC (1 mM) for 12 h. ATP production was then measured. (**C**) The mitochondrial membrane potential (ΔΨm) was determined by JC-1. Scale bar = 25 μm. (**D**) The ratios of JC-1 aggregate/monomer were analyzed by the Image J program. (**E**) Insulin-signaling assay. (**F**) Densitometry quantification of pAKT/AKT of the insulin-signaling assay in C2C12 cells. The data are presented as means ± SEM of three independent experiments. *, *p* < 0.05. T, testosterone. NAC, N-acetylcysteine; T, testosterone; Ins, insulin.

**Figure 8 ijms-23-11384-f008:**
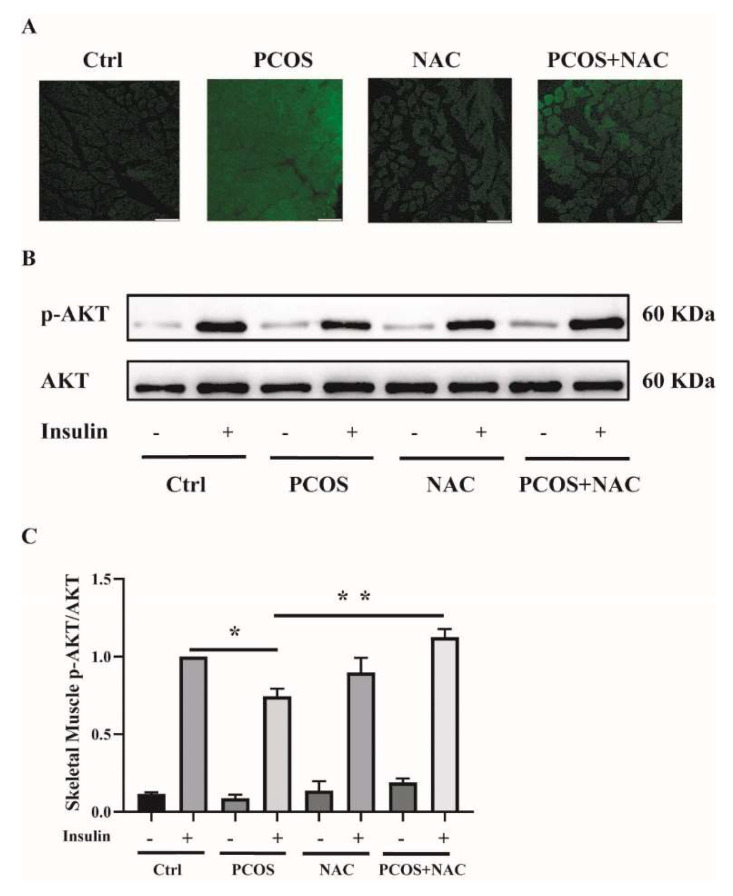
Effects of NAC intervention on mice with DHEA-induced PCOS. (**A**) Representative photomicrographs of ROS in the soleus muscles of the mice by DCFH-DA staining. Sale bar = 100 μm. (**B**) Insulin-signaling assay in the mice. (**C**) Densitometry quantification of pAKT/AKT of the insulin-signaling assay in the mice. The data are presented as means ± SEM. *, *p* < 0.05, **, *p* < 0.01, *n* = 6 per group. NAC, N-acetylcysteine.

## Data Availability

The data presented in this study are available on request from the corresponding author.

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
