# Peer review of "Oxidative Stress as a Contributor to Insulin Resistance in the Skeletal Muscles of Mice with Polycystic Ovary Syndrome"

_ijms, 2022, doi:10.3390/ijms231911384_

Round 1

Reviewer 1 Report

Comments to the author:

The manuscript entitled “Oxidative stress contributes to insulin resistance in skeletal muscle of dehydroepiandrosterone-induced polycystic ovary syndrome mice” address the role of Oxidative stress contributed to skeletal muscle IR in PCOS. Reducing ROS levels may improve insulin sensitivity of skeletal muscle in PCOS. I would like to recommend this manuscript after major revisions.

Comments

1-Please replace the low-resolution photos in Figure No. 1 with higher-quality ones.

2-Please elaborate on what makes this study different from the many others of its kind that have already been published. Example: Cell J. 2017 Apr-Jun; 19(1): 11–17. A Review on Various Uses of N-Acetyl Cysteine

3-Finally, in regard to question 3, could you please explain the mechanism by which NAC activates AKT and insulin receptors in skeletal muscle?

4-Phosphorylation of the Insulin receptor on serine 473 and threonine 1146 improves insulin sensitivity. It is recommended to assess the levels of these two proteins in cells following NAC treatment.

5-Please explain the role that IRS plays in insulin sensitivity.

Reviewer 2 Report

The authors studied the levels of oxidative stress in skeletal muscles of DHEA induced PCOS and tried to revel the molecular mechanisms responsible for it. The manuscript is interesting and has merit. I have a few comments.

1.     Introduction is not comprehensive enough. Please add some background about the mechanisms of skeletal muscle insulin resistance and its importance for overall body insulin resistance. Please add also some background about the oxidative stress and ROS.

2.      In the discussion section you state “Our data showed the increased AR expression in skeletal muscle of PCOS mice.« where was this shown in the result section?

3.     Why did you choose to analyse soleus and gastrocnemius muscles? These muscles are in mice very different by muscle fibre composition and oxidative metabolism. Where there any differences in the results between the muscles. Moreover, these muscles are also weight bearing, which may influence the results. Do you think that in non-weight bearing muscles, the results would be similar?

4.     In the figures there is stated that only soleus muscles was used, however, in the methods section you state that soleus and gastrocnemius muscles were used?

5.     Please describe in the methods section how was DCFH-DA staining and image analysis performed.

6.     Please state in the limitation that majority of mechanistic experiments were done in C2C12 cells and therefore, it is not necessarily true that the same is true for skeletal muscle cells in vivo.

7.     How was normality of the data tested for?

Round 2

Reviewer 2 Report

The authors addressed most of my objections. There are still a few minor things to consider.

1. The exact test that was used to test normality of the data using software GraphPadPrism 9 should be stated.

2. Some text is repeated in the background and discussion. This repetitions should be omitted.
